# Let’s Play the fMRI—Advantages of Gamified Paradigm in Examining the Motor Cortex of Young Children

**DOI:** 10.3390/jcm11102929

**Published:** 2022-05-22

**Authors:** Michał Podgórski, Katarzyna Matera, Łukasz Olewnik, Tomasz Puzio, Dobromiła Barańska, Piotr Grzelak

**Affiliations:** 1Department of Diagnostic Imaging and Interventional Radiology, Veteran’s Memorial Hospital, Medical University of Lodz, 90-549 Lodz, Poland; 2Department of Diagnostic Imaging, Polish Mother’s Memorial Hospital—Research Institute, 93-338 Lodz, Poland; katarzynamatera6@gmail.com (K.M.); tomekpuzio@gmail.com (T.P.); dobaranska@gmail.com (D.B.); piotr.grzelak@iczmp.edu.pl (P.G.); 3Department of Anatomical Dissection and Donation, Medical University of Lodz, 90-752 Lodz, Poland; lukasz.olewnik@umed.lodz.pl

**Keywords:** pediatrics, functional magnetic resonance imaging, gaming, paradigm, neuroimaging

## Abstract

Background: Performing functional magnetic resonance imaging (fMRI) examination is difficult when a child needs to stay awake and cooperate. Many techniques help to prepare them for the study but without modification of the examination protocol. The objective of this research was to prepare a gamified motor paradigm (“computer game”) that will improve the fMRI examination of young children. Methods: After preparing a dedicated application the fMRI examination was performed on 60 healthy children (10 girls and 10 boys in each age group of 4, 5, and 6 years old). Each child performed the gamified and a standard motor paradigm, both based on squeezing a rubber bulb. The effectiveness of squeezing were compared. Results: With the application of the gamified paradigm children completed significantly more active blocks (3.3 ± 1.4) than for the standard paradigm (2.2 ± 1.6) (*p* < 0.0001). In mixed-effects Poisson regression, age (IRR = 1.9; 95%CI: 1.5–2.5) and application of gamified paradigm (IRR = 5.6; 95%CI: 1.1–28.0) were significantly associated with more completed blocks. Conclusions: The gamified motor paradigm performed better than a standard paradigm in the fMRI examination of children between 4 and 6 years old. It allowed a significant increase in the number of completed active blocks and also better squeezing effectiveness in each block.

## 1. Introduction

Examining children in the MR scanner can be a challenge. The younger the child is, the lower the chance of success. Things are even more difficult when functional magnetic resonance imagining (fMRI) is concerned because the patient needs to cooperate to fulfill the paradigm. In general, up to 6 years of age fMRI is affected by cognitive or behavioral conditions unless performed under general anesthesia or during sleep [1]. It is well documented that pediatric subjects make on average significantly more head movements than adults [2]. Moreover, due to the relatively long time of scanning, fMRI examination is highly susceptible to head motion artifacts [3].

The above-named difficulties concern mostly motor and visuomotor paradigms. Contrary to the auditory, sensory and visual paradigms, they cannot be reliably evaluated in anesthetized or sleeping patients [1,4]. However, they are clinically relevant, e.g., in presurgical planning [5] as well as in children affected by autism [6] or epilepsy [7]. Behavioral training [8], child life specialist consultation [9], or the application of a mock scanner [10,11] have been reported to improve the quality of structural and functional MRI scans in young children. However, these techniques were not associated with tailoring the fMRI examination protocol.

New technologies also provide solutions. MR-compatible video goggles allow the projection of content (usually calm cartoons) that helps to distract awake children during MRI [12]. This is an effective but passive method, used mainly in structural scans. Quian et al. [13] first introduced a non-intrusive eye-tracking-based MR-compatible virtual reality system, where a child can ‘play a game during MRI examination’. However, it was not used to obtain particular data on brain structure/function, but to reduce the stress associated with the examination. Gaming, in a form of computerized tasks, is becoming more popular nowadays in managing pediatric patients. It was applied in diagnosing attention-deficit hyperactivity disorder [14] but we were not able to find reports on using gaming in pediatric fMRI scanning.

The study aimed to design a motor paradigm, based on the gamified application (“computer game”), that will allow performing a good-quality fMRI on children aged 4–6 years.

## 2. Materials and Methods

### 2.1. Creating the Gamified Paradigm

The “game” was prepared according to the author’s scenario to become a gamified block paradigm composed of active and passive blocks. In the active block, the participant, laying in an MRI scanner, had to squeeze the rubber bulb (clenching the bulb in a palmar opposition grasp engaging all digits of the right hand). This resulted in injecting air through the silicone tube to the receiver in the control room and finally in powering up the car that was at the start line of the track with obstacles (the patient could see it on the screen, through the mirror). During the passive block, the car started the ride and the patient observed how it traveled through the racetrack. Both active and passive blocks lasted for 30 s and were repeated 5 times (although the “game” allowed for the modification of the number of blocks and their time duration). The application also supported the adjustment of the level of difficulty. The higher the level of difficulty, the lower the increment in power for the car gained with each squeeze of the rubber bulb. A total of 50% of maximum power was enough to complete the track but higher power resulted in a more vigorous ride. Appendix A presents the sample of active and passive block performance. After finishing the paradigm, a report is available on the percentage of uploading of a power bar for the car in each active block (determined as squeezing effectiveness).

The receiver in the control room was a dedicated device compatible with the application. Specifically, it was a microphone enclosed in a sound-isolated box and adjusted to receive air portions from the silicone tube. It was connected via a USB port to the computer where the application was installed. In the receiver, there was also a module to synchronize the application with the fMRI sequence. The computer with the application (Dell G5 5587; Intel(R) Core(TM) i7-8750H CPU @ 2.20GHz 2.21 GHz; Windows 10 Pro) was connected to the screen in the scanner room via HDMI cable. In this study, the screen of the In Vivo system was used but any kind of display that can be connected to the computer can be used.

As described above, the gamified paradigm is an alternative to a standard motor paradigm, which in this case was based on squeezing the same rubber bulb. The difference was that on the screen, instead of the gameplay, were presented pictograms of “squeezing” or “not squeezing”. The standard paradigm had the same number of active and passive blocks that also lasted 30 s each.

### 2.2. Recruiting Patients

Two groups of healthy volunteers were recruited through internet advertisements:Ten adults (5 women, mean age 40 ± 7.6 years and 5 men, mean age 36 ± 6.5 years)—this group was included for the initial testing of the paradigm and examined first.Sixty children (10 girls and 10 boys in each age group of 4, 5, and 6 years old). Each child was examined by a child life specialist with the following methods:
Obtaining a developmental history from guardians;Observation of child’s behavior;The batteries of tests validated for the Polish population:
CFT 1-R-Cattell Culture Fair Intelligence Test (particular subscales);Frosting-Developmental Test of Visual Perception;Raven’s Colored Progressive Matrices.

The first two tests evaluate fluid intelligence (ability to solve novel problems, independent of any knowledge from the past) and the last assesses visual perception. These tests fit the age group of children and properly characterize the abilities of children to perform paradigms.

The inclusion criteria were: right-handedness, no history of neurological disease, and a level of development corresponding with the age of a child. Exclusion criteria comprised diseases and conditions that may disturb blood saturation with oxygen (i.e., cigarette smoking, asthma, heart failure and congenital/acquired heart defects, vascular malformations), usage of psychoactive substances and drugs influencing mental status, and standard contraindications to MRI examination (i.e., particular types of prostheses, stents, and coils for embolization).

Adults and children’s guardians obtained full information about the experiment and signed informed consent.

### 2.3. Preparation for the fMRI

Preparation aimed to familiarize participants with the gamified and standard paradigms. Firstly, the size of the rubber bulb was chosen. Three sizes were available—small, medium, and large—and all of them were animal-shaped. Then, participants learned how to “play the game” and based on their initial experiences the level of difficulty was set, so the participant by repeatedly squeezing the bulb for 30 s was able to power up the car to 70–80% of the maximum power. This range was chosen to avoid reaching 100%, which would stop a participant from squeezing before the end of the active block, and not end up below 50%, which would result in stopping the car before the finish line. Reaching the finish line is not mandatory, but we noticed that, especially for children, it is important from a motivational point of view. Afterward, participants were told how to perform a standard motor paradigm.

Additionally, children were taken to the control room to see the scanner and listen to the noise of ongoing examination through the loudspeaker.

### 2.4. fMRI Examination Protocol

Patients were placed in the MRI scanner (Achieva 3.0T, Philips, Best, The Netherlands) with ear protection pads and with a mirror fixed on the 8-channel head coil. The screen was placed behind the bore of the magnet and the view was upside down so it could be correctly seen by the patients inside the magnet. Children were accompanied by a guardian who was allowed to gently hold their lower leg. With the application of a random number generator (random.org) the order of paradigms was chosen for each patient (half of them performed the standard paradigm first and the other half started with the gamified one).

The examination started with the structural, T2-weighted scans of the head in a transverse plane (scanning parameters: TR = 3000 ms; TE = 50 ms; Flip angle 90°; matrix 512 × 512). During the structural scan, the screen behind the bore was moved aside and children watched calm cartoons provided by the Ambient Experience System (Philips, Amsterdam, The Netherlands). Afterward, the fMRI examination was performed with two echo gradient EPI (echo planar imaging) sequences (scanning parameters TR = 3000 ms; TE = 50 ms; Flip angle 90°; the interval between consecutive measurements was 3000 ms) for the standard and gamified paradigm. One run consisted of 100 image volumes for active and passive blocks preceded by 5 volumes discarded from the data evaluation to compensate for initial signal decay. Between two EPI sequences, participants had a 5 min break (staying inside the scanner with calm cartoons in the background) to give rest to the muscles of the forearm and hand. During the standard paradigm, the application was turned on but not visible for participants, who saw only pictograms. It allowed the collection of data on squeezing effectiveness because the same rubber bulb was used. The total time of examination was around 18–19 min per subject.

### 2.5. Statistical Analysis

Statistical analysis was performed with the Statistica 13 software [TIBCO Software Inc. (2017). Statistica. http://statistica.io, accessed on 10 January 2018]. According to the Shapiro–Wilk test, continuous data were not normally distributed, thus non-parametric tests were used. Separately for adults and children, numbers of correctly performed active blocks during standard and gamified paradigms were compared with the Wilcoxon signed rank test. For children, Friedman’s ANOVA with post hoc test was used to compare squeezing effectiveness during gamified and standard paradigms. Additionally, only for this group, we developed mixed-effects Poisson regression with a random intercept for each patient. In this multivariable scenario, we aimed to model a number of completed blocks based on gender, age, type of paradigm, and their pairwise interactions. Model fit was assessed with a conditional R2 parameter, using the method proposed by Nakagawa and Schielzeth [15], and the influence of parameters and their interactions were assessed using incidence rate ratios (IRR) with a 95% CI. IRR > 1 indicated that an increase in the studied parameter was associated with an increased number of completed blocks. A *p*-value of less than 0.05 was considered significant and the results are presented as the mean and standard deviation unless otherwise stated.

## 3. Results

### 3.1. Adults

All adults were able to fulfill both paradigms and there was no significant difference in the number of correctly performed active blocks between the standard 4.7 (0.5) and the gamified 4.8 (0.4) paradigm (*p* = 0.6858). These data validated the proper features of the gamified paradigm and allowed us to proceed with examining children.

### 3.2. Children

Guardians volunteered 84 children for the study; however, 11 of them were rejected after evaluation by the child life specialist (discrepancies between age and level of development that might have biased results) and another 13 were not able to undergo any of the MRI protocol steps. The remaining 60 children underwent fMRI with varying success (Appendix A).

In general, children performing the gamified paradigm completed more active blocks than for the standard paradigm (3.3 ± 1.4 vs. 2.2 ± 1.6, *p* < 0.0001). For more details, multivariate regression analysis was performed with age, gender, and types of paradigm as predictors of the number of completed active blocks. In this adjusted model, conditional R2 was 0.22, while the increase in age by 1 year (IRR = 1.90; 95%CI: 1.45–2.48) and the application of the gamified paradigm (IRR = 5.63; 95%CI: 1.13–28.00) were significantly associated with completed active blocks. Both gender (male vs. female; IRR = 0.65; 95%CI: 0.13–3.13) and all tested interactions (age and gender—IRR: 1.06 95%CI: 0.80–1.41; paradigm and age—IRR: 0.77 95%CI: 0.57–1.03; paradigm and gender—IRR: 1.23 95%CI: 0.78–1.91) had a non-significant effect.

In an analysis of squeezing effectiveness, the percentage of power bar upload significantly decreased (Figure 1) (*p* < 0.0001) with each block for both types of paradigm (Figure 1). However, according to post hoc analysis, squeezing effectiveness was significantly lower for each active block of the standard paradigm than for the corresponding block of the gamified paradigm (in all comparisons *p* < 0.0001).

## 4. Discussion

The gamified motor paradigm, in the form of a game, performed better than a standard paradigm in the fMRI examination of children between 4 and 6 years old. It increased the success rate of fMRI examinations by 70%, as a result of a significant increase in the number of completed active blocks and also by better squeezing effectiveness in each block.

There are many sufficient methods to prepare children for MRI examination [15]. Most of them aim to familiarize a child with the scanner environment and counter their fears [16]. This can be achieved by using simple equipment and play therapy (playing with a small model of an MRI scanner) [17]. However, bigger mock scanners have also been proved to be effective. De Bie et al. [10] prepared a mock scanner training protocol that allowed for the completion of the fMRI session in 36 healthy children between 4 and 6 years old (47% of the study group). In our study, with the same criteria for examination success, this rate was 56.7%. On the other hand, Yerys et al. [18] reported an 82% increase in fMRI success rate by using a mock scanner or tunnel for training in a mixed group of 4–6-year-old children (healthy and with epilepsy). However, they did not test the motor paradigm. Moreover, the design of this study (two different paradigms performed by the same participant) allowed direct improvement to be shown, without a need for comparison with historical data or only similar groups of children (like in many other studies [10]).

To avoid sedation and obtain the diagnostic quality of the MRI study, there is a trend of preparing children in a cross-curricular manner for an MRI. A combination of different methods [18] and the development of dedicated preparation protocols [19,20] can give better results than a single intervention. In this context, we find our application a perfect addition that fills the niche of the examination protocol. Other methods prepare children for the fMRI but without modification of the task that needs to be performed. Our application makes the task more attractive for a child. Furthermore, it allows for individualized adjustment of the paradigm difficulty in response to participants’ capabilities. Additionally, the registration of response (squeezing effectiveness) enables an objectified, real-time, and quantitative assessment of the task performance.

To our knowledge, this is an innovative application of gaming in fMRI. Until recently, only one research team analyzed fMRI changes in brain activity with a paradigm in a form of a video game [21,22]. However, researchers used a commercially available first-person shooter video game to assess adults’ brain functioning in response to the violence associated with killing virtual opponents.

This study has several limitations. We did not present data concerning the results of fMRI studies (e.g., how many examinations obtained diagnostic value or details concerning regions and degree of cortex activation). There was also no analysis of motion artifacts, which is especially important in pediatric, long-lasting examinations. However, this study aimed to present a gamified paradigm and was focused on the analysis of children’s performance while executing tasks. The aforementioned issues require separate analysis, which is beyond the scope of this article and will be addressed in the future. Another limitation is that only healthy children were examined. Thus, in further research children with neurological problems will be tested to confirm the usefulness of the application.

## 5. Conclusions

The gamified motor paradigm allows performing fMRI examinations in younger children than the standard paradigm, improving the completion and performance of the task.

## Figures and Tables

**Figure 1 jcm-11-02929-f001:**
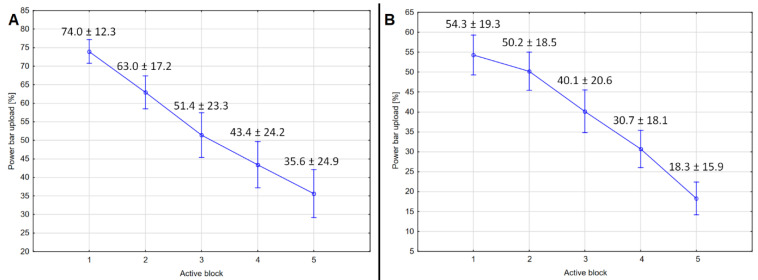
Plots indicating squeezing effectiveness for gamified (**A**) and standard (**B**) paradigms. The mean value and standard deviation for each block are presented above whiskers.

## Data Availability

Data on all software and hardware required to perform the described interactive paradigm are available upon request from the corresponding author.

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
