# Peer review of "Let’s Play the fMRI—Advantages of Gamified Paradigm in Examining the Motor Cortex of Young Children"

_jcm, 2022, doi:10.3390/jcm11102929_

Round 1

Reviewer 1 Report

The manuscript presents a very nicely conducted study and describes background, methods and results in a clear and easy- understandable manner. The "computer game" paradigm might be reasonable in clinical application and should be further examinded- as the authors suggest e.g. in adults and children with neurological disorders.

I suggest to describe the procedure of the standard paradigm earlier in the text, mabye directly after explaining the "computer game".

Please check introduction for typos. 

Author Response

Dear Reviewer,

Thank you for your precious time and valuable suggestion.

We transferred description of the standard paradigm to the section explaining design of the interactive one. We also checked the whole manuscript for topos and corrected them.

Reviewer 2 Report

In this paper the author propose the gamification of a motor task to increase the compliance of children in an MRI experiment and increase chances of successful execution of such motor tasks.

I find the topic and the proposed methods of great interest not only for MRI paediatric research and clinical work, but also in the scope of naturalistic paradigms and MRI gaming studies.

For this reason, I believe this study has great potential, that I unluckily do not find expressed in the current form of the paper. Hence, I recommend  to allow the authors to have all the time needed to improve it and reach a higher scientific impact and increase its scientific soundness. I would also encourage the authors to resubmit the revised paper, as I do believe the topic is of great interest and can serve as a proof of concept for more complex designs.

The main reason behind my decision are the suboptimal experiment design, that might introduce confounding effects that could be the cause of the presented results, and the lack of analyses and results that I believe the authors could perform and achieve with the data that they have acquired. In particular, while the current form of the paper is reminiscent of a behavioural study methods, the functional MRI data that is being collected and used as an evaluation factor allows instead for analyses proper to fMRI studies. Moreover, there are some minor points to be addressed in the current form of the paper, as well as some limitations worth discussing regarding factor and analyses that probably cannot be performed with the data presented in the paper.

The major issues that I find in this paper are related to the lack of fMRI analyses:

  • While the design seems to be sufficiently sound, the fact that the gamified task always preceded the standard task might have unintentionally introduced confounding factors in the data. In particular, the standard task might be less compelling to the subjects, since as it is presented it seems to be a less exciting version of the same task that they completed before, hence possibly increasing boredom and/or habituation effects that might explain the lower compliance to the standard task than to the gamified task. I am not aware of a way to correct for such issue, rather than acquiring more data inverting the order of the tasks. I would suggest the authors to at least perform all subsequent analyses considering only the completed trials in both tasks.
  • The authors are off to a good start by proffering the increased number of completed trials as major result to support the efficacy of the proposed protocol, using BOLD activations and responses in a small group of ROI, as evaluated from a researcher in a blind test, as a metric of successful completion. While the number of completed trials is important information and should be reported as an interesting result, BOLD activity comparison should be proffered as the major point to compare the proposed protocol to standard ones. I invite the authors to provide fMRI based result on voxelwise comparisons between activations, and to add such maps as figures for the reader to interpret such results.
  • Related to the previous point, while the blind evaluation shows the willingness to reach proper scientific standards, standardised and objective thresholds (e.g. amount of activation in ROI vs outside of ROI, lack of response, signal change %, DVARS, ...) could be used in place of a subjective evaluation. At the very least, the decision making process used for the blind evaluation should be further described and reported in the methods. This is particularly important since the paper reports individualised adjustment of the gamified task, but not of the normal task.
  • While the use of fMRI is reported, there is no description of how the data itself is (pre)processed and analysed. I would invite the authors to add this description to the methods section of the paper.
  • There is no report of the contrast and methods used to obtain figure 2. I am particularly impressed that a task presenting such rich visual stimuli does not seem to elicit activations in areas generally correlated to visual tasks, attention, and so forth. I would invite the authors to clarify why no such activations seems to be reported: was a mask used, or a contrast set up to remove them?
  • The authors report that the methods adopted in the paper are non-parametric due to the non-normal distribution of their continuous data. While they raise a valid methodological point, I invite the authors to specify what “continuous data” they are referring to. Moreover, they also adopt a linear regression, a method that normally assumes the normal distribution of the data. While it is true that there is not necessarily a non-parametric equivalent for linear regression, no attempt of normalising the distribution is carried out, nor alternative methods (such as bootstrapping) are considered. I would invite the authors to clarify such points or to adopt a coherent type of analyses throughout the study.
  • The authors report the effects of age, gender, and paradigm on the activation of predicted brain regions (table 2). However, they do not report any (lack of) interaction effect. Did the authors analyse each variable on its own? If so, I would suggest to instead use two- or three-ways ANOVAs to be able to report the interaction effects between independent variables. Otherwise, I would invite the authors to report all effects (or lack of). The same table (table 2) should be revised, since the percentages in columns 3 and 4 do not seem to match the associated numbers.
  • The authors report that the multivariate regression analysis predicted 0.57% of variance (line 200). Please check that this is a true result and not a typo: if the model explains less than 1% of the variance of the data, it would mean that the described effects, while statistically significant, could be ignored.
  • The authors refer to the BOLD contrasts as having “diagnostic value” or “resembling clinical conditions during everyday routine”. However, no citation or description of such values and conditions is reported in the text. I would invite the authors to add proper citations or further describe what they intend with these statements.

Minor issues:

  • Since both the gamified and standard task are based on squeezing an object, the associated motion artefacts could be believed to be the same. However, it would be interesting to offer a comparison between the motion artefacts resulting from both tasks: is the gamified task lowering motion artefacts due to increased attention? Or it increases motion artefacts due to increase task intensity?
  • I personally find the name interactive” task for the offered protocol partially misleading, since also the standard motor task can be considered “interactive”. Instead, I would suggest using terms like “gamified, game-like, gaming, ...”.
  • Some results that are reported in the main text (e.g. table 1) could be moved to supplementary material instead.
  • There are some redundant information, or information reported twice, like in the description of the fMRI acquisition, that might lead the reader to misunderstand the protocol. I would invite the authors to review the text and remove such points.

Furthermore, I would invite the authors to collaborate with a native English speaker, a copy editor, or a writing coach to address the typos throughout the text and improve its flow and readability.

Finally, please note that I am not an expert in gaming or paediatric MRI literature, hence I cannot objectively assess the value of the cited literature. While to the extent of my knowledge (and quick literature review) the cited literature is enough to support the presented study, I believe that further reviews of the use of gaming in MRI might help in expanding the reach of this study to other types of audience that would benefit from the results presented in this paper.

Round 2

Reviewer 2 Report

Dear authors,
Thank you for your detailed reply and text adaptation.
I would like to start apologising for overlooking lines 149-151, in which you indeed wrote that half of the subject performed the gamified task first and the other half performed it last. This strategy makes the scientific design sound. I would invite you to further clarify if the amount of times the gamified task was performed first or last is the same within sex and age group, since you later take all these factors into account. If that is not the case, I would invite you to add to the supplementary materials a table showing how many subjects performed it first or last in each sex and age group.
I perceived very positively your choice of adapting the analysis method, and I understand your intention to publish a “method” or “protocol” paper. Please consider whether it would be more appropriate to change the type of paper from “Article” to “Protocol” or “Technical Note”. I will leave a note about it to the editor as well.
This change might allow you leaving aside MRI data and results for a further publication, and instead base your observations of the number of completed trials on other parameters, such as the amount of squeezes performed (I believe you would have this datum since you are measuring it in the gamified paradigm).
If you instead prefer to measure the amount of successful trials using fMRI results, I strongly invite you to describe more in detail your preprocessing and analysis of such data. You do report what software you are using to perform any preprocessing (and analysis?) of fMRI data, which is correct and often overlooked, but please consider that this piece of information alone is not sufficient for those that would like to have a general insight on the process the fMRI data went through. First of all, readers that are not acquainted with this software will not know what happened to the data, even if everything is automatised and there is only one possible workflow. Second, it is important to report which steps the preprocessing consisted in (as you describe in your letter) and in which order. Third, it is important to report what were the parameters of some of those steps. For instance, a smoothing of 3mm FWHM will change your results compared to a smoothing of 12mm FWHM or no smoothing). The “percentage of increased blood flow” that you are referring to can vary immensely, and provide very different results considering a threshold of, for example, 5% or 95%. Which one did you use? Did you use a different threshold for each subject? If so, why did you choose that threshold and not a more liberal or conservative one? Fourth, was there any contrast you set up to have the resulting bold maps, e.g. squeezing vs passive seeing?
These would all be very important steps to help the reader understand what is happening to the data, and why you have the (very convincing) results that you have. Unluckily, I still find the choice of asking a researcher to grade such maps a peculiar choice, that, without doubting the expertise and validity of the work of such researcher, ultimately leave to human decisions what results are valid and what are not, and lowers the possibility to reproduce your results further. While manual classifications made by experts are not only necessary but also desirable in some contexts, e.g. ICA component classification, the type of classification you are performing could be perfectly parametrisable with automatic tests (e.g., using a threshold for bold activation in certain areas, using (%) volume of ROI covered by activations vs activation in other areas, …). Hence, I do not completely understand the benefit of having a researcher perform this classification. Furthermore, please consider showing more activation maps, maybe as supplementary materials.
Finally, I appreciated the linguistic and grammar revision, that truly improved the quality of your manuscript. However, I would invite you to read again the manuscript and smooth the few typos, stylistic overadjustments, and repetitions left, to further decrease the chances for readers to misunderstand your text. For instance, have a look at lines 42-44, 108, 158-159 (is the TR repeated twice, or do you refer to different types of measurements?), 248, ... (the list does not report all entries that might require revision).
